# Integrative Analysis of Expression Profiles of mRNA and MicroRNA Provides Insights of Cotton Response to *Verticillium dahliae*

**DOI:** 10.3390/ijms23094702

**Published:** 2022-04-24

**Authors:** Jun Mei, Yuqing Wu, Qingqing Niu, Meng Miao, Diandian Zhang, Yanyan Zhao, Fangfang Cai, Dongliang Yu, Liping Ke, Hongjie Feng, Yuqiang Sun

**Affiliations:** 1Plant Genomics & Molecular Improvement of Colored Fiber Lab, College of Life Sciences and Medicine, Zhejiang Sci-Tech University, Hangzhou 310018, China; 15110700054@fudan.edu.cn (J.M.); wuyuqing0104@163.com (Y.W.); qingqingn0403@163.com (Q.N.); miaom@whu.edu.cn (M.M.); devah0322@163.com (D.Z.); zhaoyanyanhao@163.com (Y.Z.); caiff@zstu.edu.cn (F.C.); yudl@zstu.edu.cn (D.Y.); keliping@zstu.edu.cn (L.K.); 2State Key Laboratory of Cotton Biology, Institute of Cotton Research of Chinese Academy of Agricultural Sciences, Anyang 455000, China; fenghongjie@caas.cn

**Keywords:** cotton, *Verticillium wilt*, transcriptome, microRNAs, network

## Abstract

Cotton *Verticillium* wilt, caused by the notorious fungal phytopathogen *Verticillium dahliae* (*V.* *dahliae*), is a destructive soil-borne vascular disease and severely decreases cotton yield and quality worldwide. Transcriptional and post-transcriptional regulation of genes responsive to *V.* *dahliae* are crucial for *V.* *dahliae* tolerance in plants. However, the specific microRNAs (miRNAs) and the miRNA/target gene crosstalk involved in cotton resistance to *Verticillium* wilt remain largely limited. To investigate the roles of regulatory RNAs under *V.* *dahliae* induction in upland cotton, mRNA and small RNA libraries were constructed from mocked and infected roots of two upland cotton cultivars with the *V.* *dahliae*-sensitive cultivar Jimian 11 (J11) and the *V.* *dahliae*-tolerant cultivar Zhongzhimian 2 (Z2). A comparative transcriptome analysis revealed 8330 transcripts were differentially expressed under *V.* *dahliae* stress and associated with several specific biological processes. Moreover, small RNA sequencing identified a total of 383 miRNAs, including 330 unique conserved miRNAs and 53 novel miRNAs. Analysis of the regulatory network involved in the response to *V.* *dahliae* stress revealed 31 differentially expressed miRNA–mRNA pairs, and the up-regulation of GhmiR395 and down-regulation of GhmiR165 were possibly involved in the response to *V. dahliae* by regulating sulfur assimilation through the GhmiR395-*APS1/3* module and the establishment of the vascular pattern and secondary cell wall formation through GhmiR165-*REV* module, respectively. The integrative analysis of mRNA and miRNA expression profiles from upland cotton lays the foundation for further investigation of regulatory mechanisms of resistance to *Verticillium* wilt in cotton and other crops.

## 1. Introduction

Cotton is a worldwide crop that plays a significant role in both the agricultural and industrial economies [1]. As the largest natural textile material, cotton accounts for more than 50% of the fiber sources in the textile industry [2]. However, cotton is susceptible to a variety of biotic and abiotic stresses during its growth, which can severely restrict its productivity. Upland cotton (*Gossypium hirsutum* L.), which accounts for approximately 97% of all cultivated cotton, is sensitive to pathogens [3]. *Verticillium wilt* is one of the major cotton yield-limiting factors and is a destructive fungal disease affecting cotton production and quality worldwide [4]. Microsclerotia, present in *V. dahliae*, can survive in the soil for many years and spread rapidly, making it difficult to control once a cotton infection occurs. The most effective way to control *V. dahliae* is the development and application of resistant cultivars. Therefore, key resistance genes and regulators need to be identified and the mechanisms mediating disease resistance must be thoroughly characterized to develop *V. dahliae*-resistant cotton cultivars. 

Plants have evolved a series of sophisticated regulatory mechanisms to avoid or resist *V. dahliae* stress. Multiple genes have been identified to play critical roles in response to *V. dahliae* stress, leading to a better understanding of their associated regulatory mechanisms. The RLK suppressor of *BIR1-1* (*GbSOBIR1*) induced by *V. dahliae* can interact with and phosphorylate *GbbHLH171* and plays a critical role in cotton resistance to *V. dahliae* [5]. By maintaining fatty acid metabolism pools for jasmonic acid (JA) biosynthesis and activating the JA signaling pathway, patatin-like proteins *GhPLP2* promote cotton resistance to *V. dahliae* [6]. Cotton laccase gene *GhLAC15* enhances *Verticillium wilt* resistance via an increase in defence-induced lignification and arabinose and xylose, which are main components of lignin in the cell walls of plants [7]. *GhSWEET42*, which encodes a glucose transporter, led to a decrease in glucose content and enhanced resistance to *V. dahliae* in cotton through glucose translocation [8]. *GhWAK7A*, a wall-associated kinase, positively regulates cotton response to *V. dahliae* infections by complexing with the chitin receptors [9]. On the contrary, as a negative regulator of resistance to *V. dahliae*, calcium-dependent protein kinase *GhCPK33* interacts with and phosphorylates 12-oxophytodienoate reductase3 (*GhOPR3*) in peroxisomes leading to decreased stability of *GhOPR3*, which consequently limits JA biosynthesis [10]. The protein kinase BIN2 that interacted with JAZ proteins regulated plant endogenous JA content and influenced the expression of JA-responsive marker genes and plays a negative role in plant resistance to *V. dahliae* [11]. The enhanced resistance in *GhIAA43*-silenced cotton plants is due to the activation of salicylic acid (SA)-related defenses, and the activated defenses specifically occurred in the presence of *V. dahliae*, suggesting that *GhIAA43* is a negative regulator in cotton defense against *V. dahliae* attack [12].

In addition, miRNAs, a class of tiny non-coding RNAs, which are considered to be important regulators at the post-transcriptional level, are also efficacious to control the gene expression during *V. dahliae* induction [13]. In plants, miRNAs act on specific target mRNAs in a complete or near-perfect base-pairing manner, resulting in mRNA degradation or translation inhibition [14]. The available evidence indicates that miRNAs play influential roles in regulating cotton response to *V. dahliae.* Inoculated with the different *V. dahliae* strains in upland cotton variety KV-1, a *V. dahliae*-tolerant cultivar, thirty-seven novel miRNAs were identified after small RNA sequencing [15]. After infection with *V. dahliae* in susceptible and resistant varieties of cotton, a total of 140 conserved miRNAs and 58 novel miRNAs were identified and 107 genes targeted by 45 miRNA families were detected via small RNA sequencing and degradome sequencing [16]. Recently, the regulatory network model of GhmiR477-*GhCBP60A* interactions and GhmiR164-*GhNAC100* interactions that enhanced the resistance to *V. dahliae* has been elucidated in detail [17,18]. In addition, the GhmiR397-*GhLAC4* module that was identified as a negative regulator of resistance to *V. dahliae* was also resolved [19]. Although many cotton miRNAs have been identified in previous research, only a small portion have been experimentally validated and the role of miRNAs in the regulation of *V. dahliae* defense responses remains unclear. 

Nevertheless, previous studies have mainly focused on differential expression analysis of *V. dahliae*-induced genes at a single level, such as transcriptome or miRNAome analysis in cotton. To date, there have been few reports on integrative analysis of mRNAs and miRNA expression that reveal the complicated network involved in regulating the response to *V. dahliae* infection in resistant and susceptible cotton genotypes. This information, including miRNA–mRNA regulatory mechanisms associated with upland cotton resistant to *Verticillium wilt*, is still limited. Therefore, a genome-wide profiling of both mRNAs and miRNAs may shed more light on the underlying regulatory mechanisms. To further explore the key factors in response to *V. dahliae* resistance and to obtain a better understanding of the molecular basis of the *V. dahliae* stress response in cotton, mRNA and miRNA expression were simultaneously profiled. Systematic investigations of the regulatory networks were performed under *V. dahliae* stress using high-throughput sequencing. Clusters of differentially expressed genes that might respond to *V. dahliae* tolerance in cotton were identified to be related to various pathways. Furthermore, quantitative reverse transcription-PCR (qRT-PCR) analysis was carried out to validate the expression patterns of several important candidate genes. The transcriptome and miRNA data obtained should help reveal critical genes and miRNAs underlying the *V. dahliae* defense responses in cotton and provide new insights into dynamic antifungal regulation in plants.

## 2. Results

### 2.1. Differential Expression of mRNAs between J11 and Z2 Response to V. dahliae Stress

The *V. dahliae*-sensitive cultivar J11 (*Gossypium hirsutum* L.) and *V. dahliae*-tolerant cultivar Z2 (*Gossypium hirsutum* L.) (Appendix A) provided by the Institute of Cotton Research of Chinese Academy of Agricultural Sciences were used to test mRNA and miRNA responses to *V. dahliae* stress. To identify the mRNAs involved in plant responses to *V. dahliae* stress, 12 RNA libraries for three biological replicates constructed from mock and *V. dahliae*-inoculated J11 and Z2 were sequenced. After removing the reads with linkers and low-quality reads, 18.9–24.6 million clean reads with Q30 ranged from 92.1–93.6% were obtained (Appendix A). Genes with differential expression between susceptible and resistance cottons (cutoff fold change ≥ 2 and *p*-value ≤ 0.05) were defined (Figure 1, Appendix A). A total of 6701 and 1629 DEGs with ≥2-fold change were detected in J11 and Z2, respectively, including 4252 up-regulated and 2449 down-regulated DEGs in J11 and 1213 up-regulated and 416 down-regulated DEGs in Z2 (Figure 1A,B). In particular, the number of up-regulated DEGs was significantly higher than that of down-regulated DEGs and the number of DEGs in J11 was significantly more than those in Z2, almost 3.5 times the number of up- and 1.5 times down-regulated DEGs in Z2 (Figure 1C). Among the DEGs, 816 unigenes were co-upregulated in both J11 and Z2, whereas 208 unigenes were co-downregulated in both J11 and Z2 (Figure 1D, Appendix A). These results likely reflect fungal growth in J11 and massive induction and activation of stress responsive genes, whereas Z2 could hardly be infected. 

### 2.2. Functional Classifications of DEGs in Response to V. dahliae Stress

For functional classifications of DEGs, GO enrichment analysis was conducted to analyze the possible biological functions of DEGs responsive to *V. dahliae* infection in the two cotton lines. Enrichment analysis revealed specific biological processes and metabolic pathways that were differentially represented in *V. dahliae*-inoculated cotton genotypes (Figure 2A). There was a higher proportion of upregulated oxidoreductase activity and peroxidase activity in Z2 compared to J11, thus improving *V. dahliae* tolerance through the regulation of antioxidant ability (Figure 2A). In contrast, the increased proportion of DNA binding transcription factor activity, signal transduction, biotic stimulus responses, and stress responses in J11 were significantly higher than those in Z2 (Figure 2A). These results revealed that diverse processes are involved in regulating cotton responses to *V. dahliae* and overall J11 exhibited a more pronounced disease response. Additionally, 1024 co-DEGs were selected for GO analysis. Further analysis revealed that biological processes related to the kinesin complex, microtubule binding, microtubule motor activity, iron ion homeostasis, iron ion transport and ferric iron binding were significantly overrepresented (Figure 2B). Notably, biological processes related to biotic stimulus responses were also identified (Figure 2B). Overall, the RNA sequencing data suggested that the early upregulation of selected genes from oxidoreductase activity, peroxidase activity, microtubule activity, and the downregulation of genes involved in iron ion homeostasis, iron ion transport, and ferric iron binding may contribute to resistance. 

### 2.3. DEGs of WRKY Transcription Factors in Response to V. dahliae Stress

Increasing evidence shows that plant transcription factors partake in defense against pathogen infection [4]. Among the DEGs, 251 transcription factor families such as WRKY, C2H2, MYB-related, bHLH, and bZIP were identified to be upregulated or downregulated in J11. Compared with other transcription factor families, the WRKY family accounted for the largest proportion (18.3%) of differential expressions indicated that *WRKY* genes primarily responded to *V. dahliae* stress in J11 (Figure 3). WRKY proteins are plant-specific transcription factors known for their function in plant defense by acting downstream of many immune response pathways [20]. A total of 60 *WRKY* genes were induced among the DEGs, including 46 *WRKY* genes in J11 and 14 in Z2 (Figure 3, Appendix A). In J11, upregulated and downregulated *WRKY* genes were divided equally at 24 h post infection (hpi) (Figure 3A). In contrast, almost all *WRKY* genes were upregulated, except a *WRKY70* gene (Gh_A02G0029) and a *WRKY51* gene (Gh_A02G1301) downregulated at 24 hpi in Z2 (Figure 3B). Interestingly, another *WRKY70* gene (Gh_D05G2642) was upregulated in J11 (Figure 3A). Even though *WRKY40* has been implicated in resistance to both fungal and bacterial pathogens [21], three upregulations of *WRKY40* homologs were found in both J11 and Z2. In addition, seven *WRKY6* homologs were identified in J11, of which three were upregulated and four were downregulated (Figure 3). In contrast, both *WRKY6s* identified in Z2 were upregulated. These results were indicative of a potential positive regulatory role for these up-regulated *WRKY* genes in cotton defense responses to *V. dahliae* stress (Figure 3).

### 2.4. Identification of Cotton miRNAs in Response to V. dahliae Stress

To identify the miRNAs involved in plant responses to *V. dahliae* stress, 12 small RNA libraries for three biological replicates constructed from mock and *V. dahliae* inoculated J11 and Z2 were sequenced. After removal of the adapter sequence, low-quality reads, and <15 nt reads, 9.8–16.7 million clean reads were obtained in these libraries and sequences with lengths ranging from 17 to 30 nt were selected for the following analysis (Appendix A). A similar distribution of reads appeared in all samples, and the majority of the small RNAs were concentrated in the range of 21–24 nt, with 24 nt being the most abundant, followed by the 21 and 22 nt categories across all of the libraries, which suggests that no significant degradation of small RNAs occurred during *V. dahliae* stress (Appendix A). These small RNAs were searched against miRBase version 21 and against 330 conserved miRNAs reported in previous studies (Figure 4, Appendix A). In addition, de novo predictions have been performed that identified another 53 miRNA candidates (Figure 4, Appendix A).

Comparisons with the mock revealed 30 and 36 DEMs in J11 and Z2, respectively (cutoff fold change ≥ 1 and *p*-value ≤ 0.05). Among these genes, 24 significantly upregulated and 6 downregulated miRNAs were found in J11, where novel-miRNA-12, novel-miRNA-42, novel-miRNA-45 and novel-miRNA-52 were upregulated and novel-miRNA-40 was downregulated (Figure 4A). In Z2, 33 miRNAs were significantly upregulated and 3 were downregulated. Among them, novel-miRNA-33 and novel-miRNA-52 were upregulated and novel-miRNA-32, novel-miRNA-40, and novel-miRNA-42 were downregulated (Figure 4B). There were eight co-upregulated miRNAs and one co-downregulated miRNA in both J11 and Z2 (Figure 4). Furthermore, over 80% of the miRNAs were upregulated, indicating that increased expression of miRNAs might play a more crucial role than the decreased expression of miRNAs in cotton defense responses to *V. dahliae* stress. Notably, the number of upregulated DEMs in Z2 was significantly higher than those in J11, indicating its superiority in terms of disease resistance (Figure 4).

### 2.5. Target Prediction and Construction of a Regulatory and Interaction Network

DEM target prediction and GO annotation analyses were performed to characterize the regulatory roles of miRNAs in response to *V. dahliae*. A total of 12,097 and 6819 target unigenes for DEMs were identified in J11 and Z2, respectively. For the target unigenes found in J11, among the GO terms identified in the biological process, two significantly enriched categories were “response to hormone” and “lignin catabolic process” (Appendix A). For the target unigenes found in Z2, the biological process of “protein ubiquitination”, “response to hormone”, and “lignin catabolic process” were significantly enriched (Appendix A).

To establish the regulatory network of miRNA–mRNA involved in the response to *V. dahliae*, the potential targets of DEMs were analyzed from DEGs in the transcriptome. After excluding miRNA–mRNA modules with the same expression pattern, a total of 23 miRNA–mRNA pairs were found in J11, involving 8 DEMs and 20 DEGs, showing antagonistic regulatory patterns involving up-regulated miRNAs and downregulated mRNAs (Figure 5A, Appendix A). Among these pairs, 11 pairs involved downregulated miRNAs and upregulated mRNAs; 12 pairs involved upregulated miRNAs and downregulated mRNAs (Figure 5A, Appendix A). In Z2, a total of eight miRNA-mRNA pairs were found, involving three DEMs and eight DEGs (Figure 5B, Appendix A). Among these pairs, seven pairs involved down-regulated miRNAs and up-regulated mRNAs; only one pair involved up-regulated miRNAs and down-regulated mRNAs (Figure 5B, Appendix A). The affected pathways in the miRNA-mRNA regulatory network included the “response to hormone” and “lignin catabolic process” pathways, many of which have been described previously. In depth, the GhmiR395-*APS1/3* module (upregulated miRNAs) and GhmiR165-*REV* module (downregulated miRNAs) had higher expression levels relative to other miRNAs and were obviously induced after inoculation with *V. dahliae*. These were validated by 5′ RLM-RACE and qRT-PCR, which suggested that the selected modules were possibly involved in the response to *V. dahliae* stress (Figure 6A,B).

### 2.6. Validation of Gene Expression by qRT-PCR

To verify the reliability of the RNA sequencing results, the relative expression levels analysis of selected genes was investigated with qRT-PCR. According to the results obtained from qRT-PCR, the expression pattern of eight selected mRNAs was largely similar with our sequencing data, including four upregulated and four downregulated mRNAs (Appendix A). As for miRNAs, stem-loop qRT-PCR was used to determine the miRNA expression levels of eight miRNAs with three biological replicates. The values of relative expression level of GhmiR160b, GhmiR164, GhmiR166d, GhmiR395, GhmiR398a and GhmiR403 were all positive, while GhmiR165 and novel-miRNA-40 were negative (Appendix A), which were closely matched with sequencing data. Our present results showed that the expression profiles of those miRNAs and mRNAs were reliable to be used to investigate V. dahliae-induced transcriptional changes in cotton.

## 3. Discussion

*Verticillium* wilt of cotton caused by the notorious fungal pathogen *V. dahliae* is a destructive soil-borne vascular disease that severely decreases cotton yield and quality worldwide. The *Verticillium* pathogen typically invades and colonizes the roots of plants, and then spreads acropetally through the vascular tissue of the plant, resulting in the primary symptoms of necrotic areas on leaves, yellowing of leaves, wilting, and discoloration of vascular tissues [4]. Unfortunately, upland cotton (*G. hirsutum* L.), the main species cultivated on a large scale, is sensitive to *V. dahliae*. However, the specific molecular and genetic components that mediate resistance to the pathogen and the plant immune response pathways remain largely unknown. Two contrasting upland cotton genotypes (highly resistant and susceptible), with distinct defense responses, were used in the global expression profiling experiment described here to investigate the roles of regulatory RNAs in *V. dahliae* induction. Genome-wide mRNA and miRNA profiling data provide genes regulated during pathogen infection, which can be pursued through genetic studies or used as markers for tracking the activation of immune responses.

Comparative transcriptomic analysis revealed that the number of DEGs in J11 was significantly higher than that in Z2, revealing a stronger response induced in the susceptible genotype (Figure 1D). In addition, the number of upregulated DEGs was significantly higher than that of downregulated DEGs in both J11 and Z2, suggesting that early upregulation of selected genes contributes to disease resistance (Figure 1D). The accumulation of reactive oxygen species (ROS) is a classical immune response that affects resistance to pathogens [22]. Changes in the redox state of glutathione and the accumulation of ROS in the cytosol during biotic stress can initiate the activation of defense genes in the nucleus through pathways that involve many plant hormones [23]. Interestingly, there was a greater proportion of upregulated DEGs that encode proteins involved in oxidoreductase activity and peroxidase activity, such as Gh_D08G0426, Gh_A10G2290, Gh_D10G0605, Gh_D04G1116 and Gh_D12G0699. These specific proteins may encode oxidoreductases and peroxidases that affect the synthesis of glutathione and ROS directly or indirectly in Z2 compared with J11, thus exhibiting a remarkable level of resistance to *V. dahlia* infection in Z2 (Figure 2A). In contrast, the increased proportion of DNA binding transcription factor activity, signal transduction, biotic stimulus response and stress response in J11 were significantly higher than those in Z2, likely due to massive changes in disease symptoms and fungal growth in J11, but not in Z2. These results were consistent with the extent of disease symptoms and fungal growth in J11 and Z2 (Figure 2A). Lots of evidence indicates that microtubule disruption is associated with plant defense responses and resistance [24]. During pathogen infections, plant microtubules are commandeered by the pathogen for intra and intercellular movement, as well as for interhost transmission [25]. With regard to the co-DEG functional analysis, our results show that the major proportion of upregulated DEGs were within the functional categories of microtubule activities, including microtubule binding, kinesin complex, and microtubule motor activity, indicating that active microtubule movements in cotton respond to *V. dahliae* infection (Figure 2B). Therefore, the biochemical and physical mechanisms by which microtubule activity responds during *V. dahliae* infection should be elucidated. The formation of localized cell wall appositions, oxidative bursts, and production of pathogenesis-related proteins are hallmarks of plant defense responses, and iron is a central mediator that links these three phenomena [26]. In response to a pathogen attack, the bulk secretion of Fe^3+^ is deposited at cell wall appositions, where it accumulates and leads to intracellular iron depletion, which promotes the transcription of pathogenesis-related genes [26]. Additionally, some plant pathogens use siderophores to acquire iron in the host and trigger host immunity through the perturbation of heavy-metal homeostasis [27]. Notably, a large proportion of the downregulated co-DEGs were within the functional categories of iron ion homeostasis, iron ion transport and ferric iron binding. For instance, some genes such as Gh_D09G1582, Gh_A09G2418, Gh_A10G2147, Gh_D10G2394 appeared to encode the ferritin-3 protein, which likely impacts the resistance and severity of disease symptoms (Figure 2B, Appendix A). These results provide valuable insights into the multiple molecular pathways that control the cotton stress response to *V. dahliae* (Figure 7).

The WRKY gene family is one of the largest families of transcription factors in higher plants and has been shown to play an important role in plant defense responses to a variety of pathogens [20]. *WRKY* genes may activate or suppress the expression of resistance genes directly or interact with other transcription factors to regulate plant defense responses [20]. WRKY70 is considered a repressor of JA-responsive genes [28]. Notably, Gh_A02G0029 (*WRKY70*) was downregulated in Z2, but a *WRKY70* gene (Gh_D05G2642) was upregulated in J11, suggesting a negative regulatory role in response to *V. dahliae* infection through inhibition of the JA signaling pathway (Figure 3, Appendix A). *F.oxysporum*-induced *GhWRKY40* plays a negative role in disease resistance in cotton by disrupting the SA-mediated defense pathway [29]. In our study, the upregulation of *WRKY40* homologs were found in both J11 and Z2. It is likely that *WRKY40* responds to *V. dahliae* infection by activating the SA signaling pathway (Figure 3, Appendix A). Additionally, *GhWRKY6* was induced by drought and salt stress and acted as a negative regulator during both drought and salt stress [30]. Among the DEGs, four downregulated *WRKY6* homologs were identified in J11. In contrast, both *WRKY6* genes identified in Z2 were upregulated, indicating that they contribute to resistance in Z2 (Figure 3, Appendix A). This result revealed that WRKY6s are also induced by *V. dahlia* and may have opposing functions in the fight against *V. dahliae* infection. Finally, we found other WRKY homologs with different expression patterns induced by *V. dahliae*, which implied *WRKY* genes may play both negative and positive regulatory roles between susceptible and resistant cotton cultivars (Figure 7).

It has become clear that miRNAs regulate gene expression mainly by cleaving target mRNAs or through transcriptional inhibition at the post-transcriptional level and play a crucial role in coordinating plant–pathogen interactions. The up-regulation of miR159, miR160, miR164, miR166 and miR167 was consistent with results reported in *Arabidopsis* and cotton [15,16,31]. Although miR398 may be involved in immune responses with an inverse regulatory mode of action in bacterial and fungal pathogen species [32], miR398 homologs were significantly upregulated in both J11 and Z2, indicating that they may participate in the *V. dahliae* infection response (Figure 4, Appendix A). In addition, co-upregulated novel-miRNA-52 and co-downregulated novel-miRNA-40 in both resistant and susceptible lines indicated that these novel miRNAs may play different roles in the response to *V. dahliae* infection (Figure 4, Appendix A). It is worth noting that novel miRNA-42, which had opposing inducible expression patterns in resistant and susceptible lines, may serve as a potential molecular target to improve *V. dahliae* resistance in cotton (Figure 4, Appendix A).

In cotton, cleavage of *GhNAC100* mRNA by GhmiR164 leads to degradation of *GhNAC100*, thereby enhancing its resistance to *V. dahliae* [18]. GhmiR397 cleaved the *GhLAC4* transcript and was identified as a negative regulator of lignin biosynthesis that improves plant resistance to infection by *V. dahliae* [19]. According to the established miRNA–mRNA interactions, upregulation of miR164 and miR397 was found in J11 and Z2. However, *GhNAC100* targeted by miR164 and *GhLAC4* targeted by miR397 were also upregulated. These two modules failed to show antagonistic regulatory patterns, likely due to the later stage of induction of fungal-infected cotton when *V. dahliae* colonized the xylem vessels. Sufficient levels of sulfur in soils confer the optimal plant uptake of inorganic sulfate salts, a prerequisite for sulfur-containing defense compound concentrations required for plant disease resistance responses [33]. In rice, miR395 targets and suppresses the expression of the ATP sulfurylase gene *OsAPS1*, which functions in sulfate assimilation, to promote sulfate accumulation, resulting in broad-spectrum bacterial resistance [34]. As a specific colonizer of plant xylem vessels, *V. dahliae* encodes a complex array of cell-wall-degrading enzymes that invade plant vascular tissue [4]. A group of HD-ZIP III transcription factors including *REV*, *PHB* and *PHV,* whose expression is post-transcriptionally regulated by miR165/166, play key roles in the establishment of the vascular pattern and secondary cell wall formation in plants [35]. It is likely that plants use miR165-HD-ZIP III transcription factor modules to prevent pathogen invasion by altering the development of vascular tissue and cell wall formation. In this study, modules such as GhmiR395-*APS1*/*APS3* and GhmiR165-*REV* validated by 5′ RLM-RACE did work in cotton; therefore, it remains worthwhile to explore whether these miRNAs and their targets are involved in the response to *V. dahliae* infection in cotton (Figure 7). Moreover, examination of the novel regulatory networks consisting of miRNAs and their corresponding targets will help us to better understand the regulatory mechanisms of stress responses to *V. dahliae* infection (Figure 5, Appendix A).

## 4. Materials and Methods

### 4.1. Plant Materials and Pathogen Treatment

The *V. dahliae*-sensitive cultivar J11 and *V. dahliae*-tolerant cultivar Z2 (Appendix A) [36] were used to test mRNA and miRNA response to *V. dahliae* stress. Cotton seeds were germinated on wet gauze in a Petri dish at 28 °C and cultivated in a growth chamber with the same conditions of 25 °C/18 °C cycle under a 16 h light/8 h dark cycle. Vd080, a highly pathogenic *V. dahliae* strain [36], was confirmed and then cultured in potato dextrose broth (PDA) in plates until concentration was adjusted to 10^6^ spores/mL to inoculate cotton roots for 24 h. As a mock-inoculation control, roots were inoculated with empty PDA. For material harvest, the roots from six different plants in each treatment were mixed separately, replicated three times and immediately plunged in liquid nitrogen and stored at −80 °C for RNA isolation.

### 4.2. RNA Preparation and Sequencing

Roots from mock and *V. dahliae*-treated cotton were used for RNA library construction and deep sequencing analyses according to [37]. Total RNA was isolated by using the RNA reagent (Invitrogen, Carlsbad, CA, USA). The quantity and quality of the isolated total RNA were assessed using a NanoDrop OneC Spectrophotometer. For transcriptome sequencing, the enriched mRNAs were purified from the total RNA with magnetic beads attached to oligo (dT) and the strand-specific libraries were sequenced on Illumina Hiseq2500 platform at Novogene (Beijing, China) with pair-end strategy (2 × 150 bp). Moreover, small RNA libraries were constructed using the Small RNA Sample Prep Kit (Illumina, San Diego, CA, USA) and sequencing was performed on Hiseq platform at Novogene (Beijing, China) with single end strategy (50 bp).

### 4.3. Identification of *DEGs* *and* *DEMs*

Clean data from RNA sequencing were aligned to the reference genome (NAU, v1.1, downloaded from CottonGen) by TopHat (v2.1.0). HTSeq was used to calculate the number of short reads aligned to the characterized gene loci, and DESeq2 was then used to identify the DEGs (cutoff fold change ≥ 2 and *p*-value ≤ 0.05). Quality control of the small RNA sequencing data was carried out by using FASTX-Toolkit (http://hannonlab.cshl.edu/fastx_toolkit/, accessed on 16 February 2022). Conserved miRNAs identification and de novo prediction of miRNAs were performed according to [37]. Predicted miRNAs that derived from tRNA, rRNA and snoRNA were removed before further analysis. Expression of the small RNAs was calculated by RPM (reads per million mapped reads) using in-house developed Perl scripts. DEMs were also identified by DESeq2 (cutoff fold change ≥ 1 and *p*-value ≤ 0.05). Enrichment of gene ontology (GO) terms and pathways in DEGs was estimated by chi-square testing. Only the GO terms referring to more than three genes were retained, and the redundancy of GO terms was removed through the online tool REViGO with default settings [38].

### 4.4. Integrated Analysis of mRNA and miRNA Sequencing

The prediction of miRNA target genes was performed using Target Finder software. The miRNA–mRNA pairs with a reverse expression correlation relationship were collected to construct the miRNA–mRNA regulatory network. Gene ontology enrichment analysis (http://www.geneontology.org, accessed on 24 March 2022) was carried out for allocating genes to different functional categories and predicting their biological functions, respectively [38].

### 4.5. Expression Analyses of mRNA and miRNA

Reverse transcription was performed using the PrimeScript™RT reagent kit with gDNA Eraser (Takara). qRT-PCR was performed using the TB Green^®^ *Premix Ex Taq*™ (Takara) on the QuantStudio 3 Applied Biosystem and quantified using the ∆∆C_T_ method. *Gh**UBQ7* and U6 snRNA were used as internal controls for mRNA and miRNA, respectively [37]. The specificity of the qPCR reactions was confirmed by melting curve analysis of amplified products. Primers used for qPCR are listed in Appendix A. The comparisons between samples were performed using the Student’s *t*-test. Two-tailed *p*-values of less than 0.05 were considered to be statistically significant. SPSS Statistics 19.0 was used to conduct the analysis [37].

### 4.6. 5’-RNA Ligase-Mediated Rapid Amplification of cDNA Ends (5’ RLM-RACE)

The RNA of a mixture of cotton roots was extracted. The cleavage sites of targets were identified through GeneRacer Kit (Invitrogen, CA, USA) according to the manufacturer’s guidelines. The PCR products purified by gel were subcloned into pMD18-T vectors (Takara). The clone sequences were analyzed to map the cleavage sites. The gene-specific primers were listed in Appendix A.

## 5. Conclusions

In this study, mRNA libraries and small RNA libraries were integrated to systematically investigate the roles of regulatory RNAs under *V. dahliae* induction in upland cotton. Biological processes such as antioxidant ability, microtubule activities and iron ion homeostasis were thought to be important in response to *V. dahliae* infection. Our findings suggested that the differential regulation and expression of *WRKYs* play both negative and positive regulatory roles in *V. dahliae* stress responses. In addition, our analysis also identified miRNAs that may play important roles in the response to *V. dahliae* stress. Many of the miRNAs and their target genes were involved in the regulation of “protein ubiquitination”, “response to hormone” and “lignin catabolic process”. Moreover, up-regulation of GhmiR395 and down-regulation of GhmiR165 were possibly involved in response to *V. dahliae* by regulating sulfur assimilation through the GhmiR395-*APS1/3* module and the establishment of the vascular pattern and secondary cell wall formation through the GhmiR165-*REV* module, respectively. In summary, this work found some key genes/modules involved in the response to *V. dahliae* stress in cotton that provided several candidate targets of genetic modification for further use in resistance to *Verticillium wilt.*

## Figures and Tables

**Figure 1 ijms-23-04702-f001:**
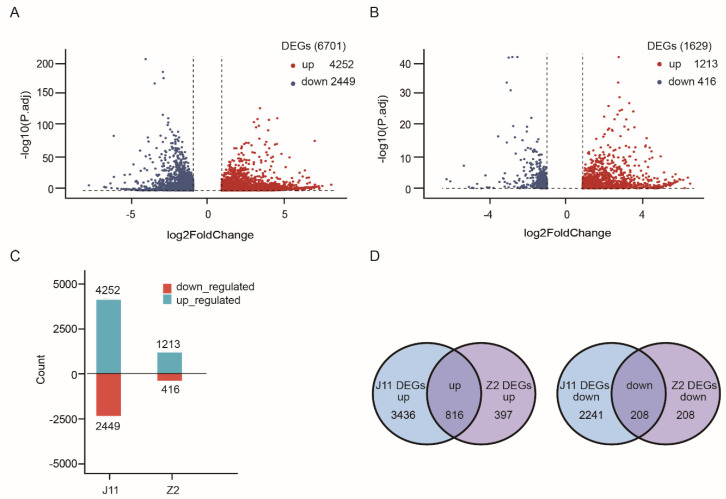
Global evaluation of the RNA sequencing data in upland cotton lines J11 and Z2. (**A**) Volcanic plot of the differential mRNA expression in J11. (**B**) Volcanic plot of the differential mRNA expression in Z2. (**C**) Counts of down- and up-regulated genes in J11 and Z2, respectively. (**D**) Venn diagram of the differential mRNA expression between J11 and Z2.

**Figure 2 ijms-23-04702-f002:**
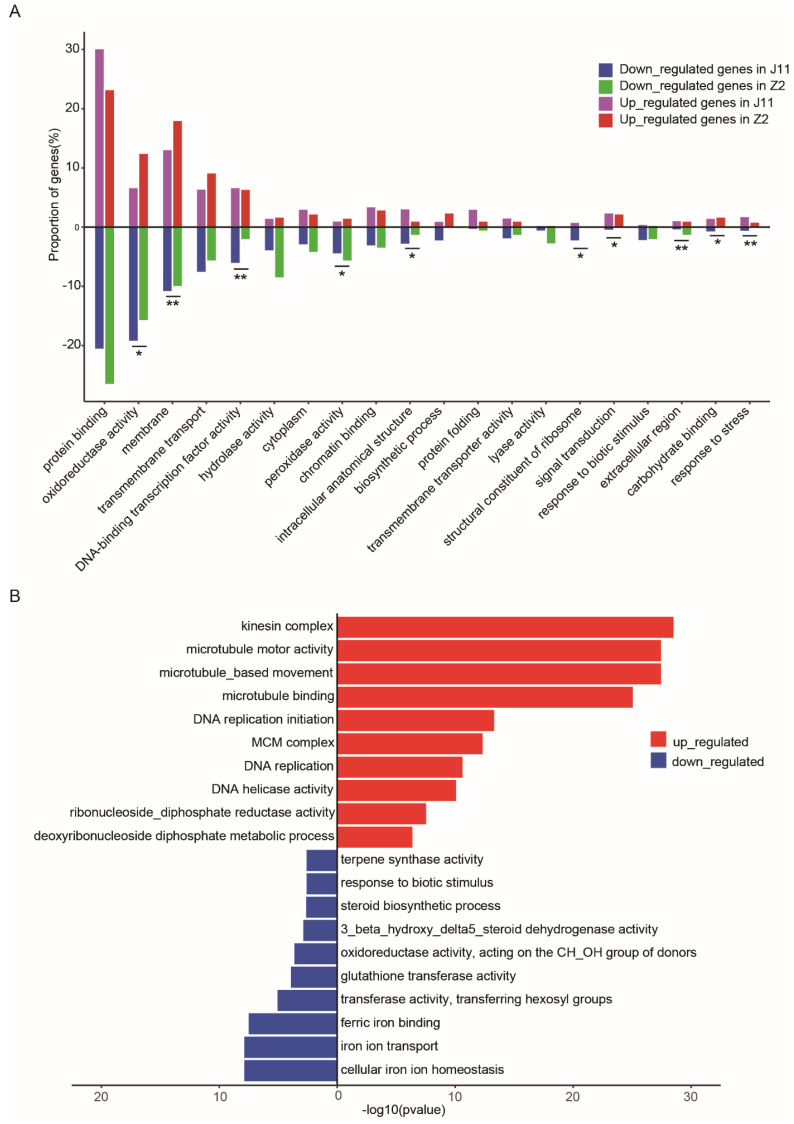
Overview of genes differentially expressed in response to *V. dahliae* stress in J11 and Z2. (**A**). Functional categories of DEGs after *V. dahliae* inoculation. The percentages of upregulated and downregulated genes are shown for the selected overrepresented functional categories. The significance of the difference between two accessions (chi-squre test) are analyzed with “*” (*p* < 0.05) or “**” (*p* < 0.01). (**B**). GO enrichment analysis of the co-upregulated and co-downregulated unigenes with the top10 are shown.

**Figure 3 ijms-23-04702-f003:**
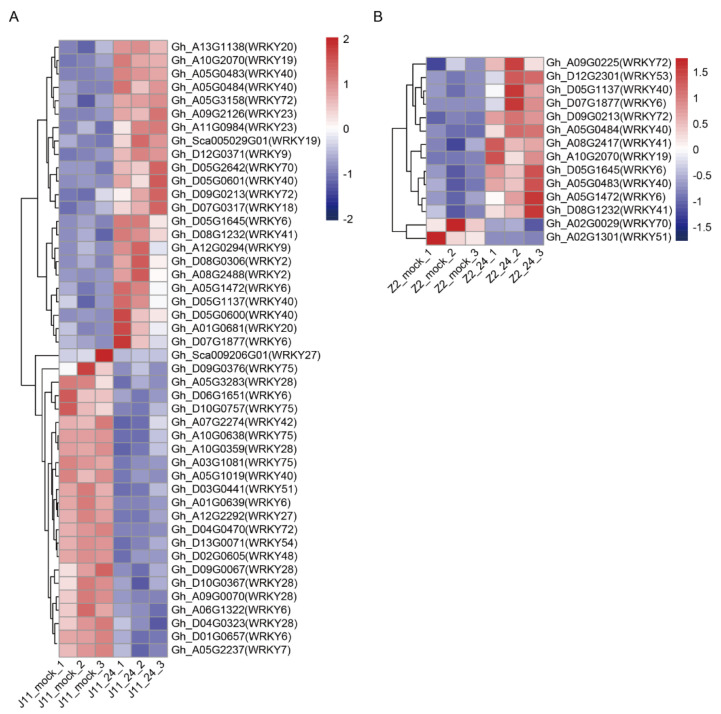
Expression profiling of *WRKYs* in response to *V. dahliae* stress. Hierarchical clustering of differentially expressed *WRKYs* in J11 (**A**) and Z2 (**B**). Red and blue colors show upregulation and downregulation, respectively. The original expression values of the *WRKYs* were normalized using Z-score. The signal intensity ranges from −2 to 2, as the corresponding color also changes from blue to red.

**Figure 4 ijms-23-04702-f004:**
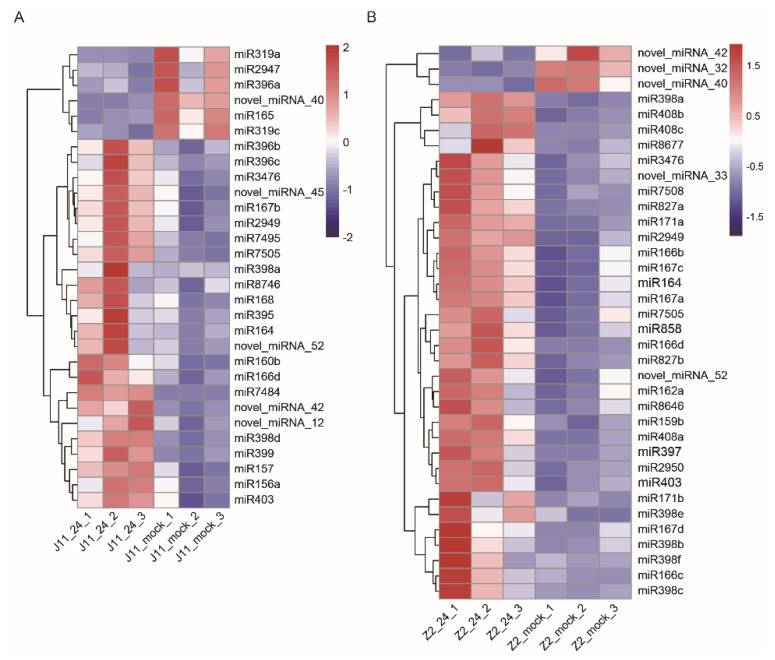
Expression profiling of miRNAs in response to *V. dahliae* stress in J11 (**A**) and Z2 (**B**). Hierarchical clustering of differentially expressed miRNAs in J11 (**A**) and Z2 (**B**). Red and blue colors show upregulation and downregulation, respectively. The original expression values of the miRNAs were normalized using Z-score. The signal intensity ranges from −2 to 2 and −1.5–1.5 as the corresponding color also changes from blue to red.

**Figure 5 ijms-23-04702-f005:**
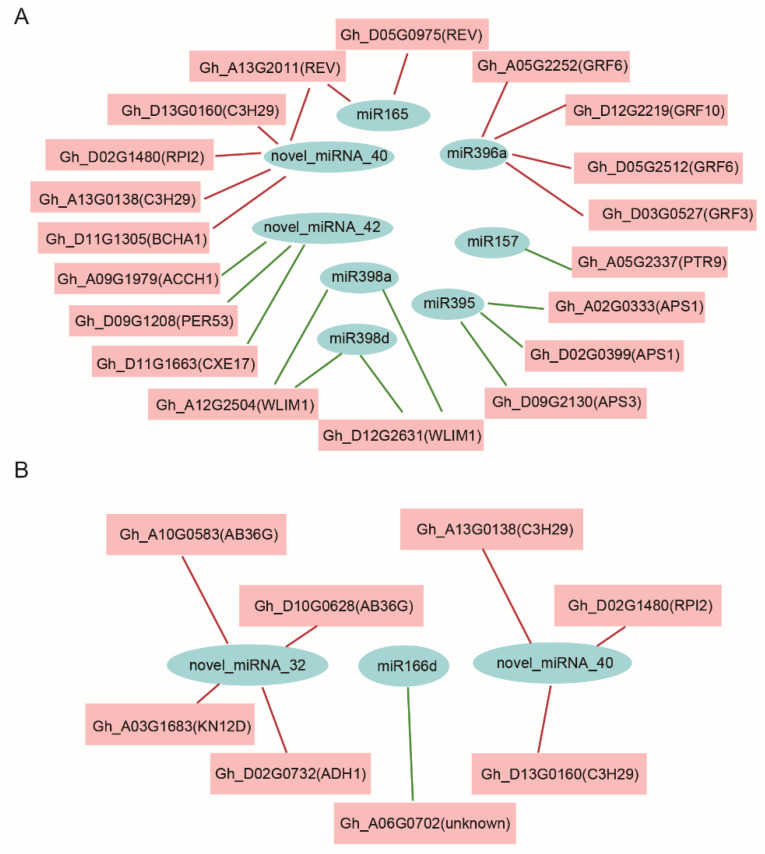
Regulatory network between miRNAs and target mRNAs associated with *V. dahliae* stress in J11 (**A**) and Z2 (**B**). Red lines represent downregulated miRNAs and upregulated mRNA pairs, green lines represent upregulated miRNAs and downregulated mRNA pairs.

**Figure 6 ijms-23-04702-f006:**
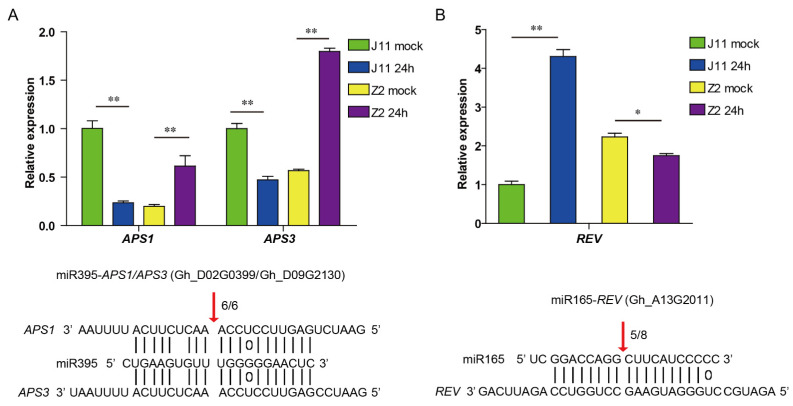
GhmiR395-*APS1/3* and GhmiR165-*REV* module were validated by qRT-PCR and 5′ RLM-RACE. (**A**). qRT-PCR was performed to determine the expression levels of *GhAPS1/APS3/REV.*
*GhUBQ7* was used as an internal control and data are means ± SD from three biological replicates. * and ** indicates a significant difference at *p* < 0.05 and *p* < 0.01 compared with mock using a two-tailed Student’s *t*-test. (**B**). The cleavage sites of *GhAPS1/APS3/REV* mRNA were determined by the 5′ RLM-RACE method. Arrow indicates the 5′ terminus of miRNA-guided cleavage products, as identified by 5′-RACE, with the frequency of clones (6/6 and 5/8) are shown.

**Figure 7 ijms-23-04702-f007:**
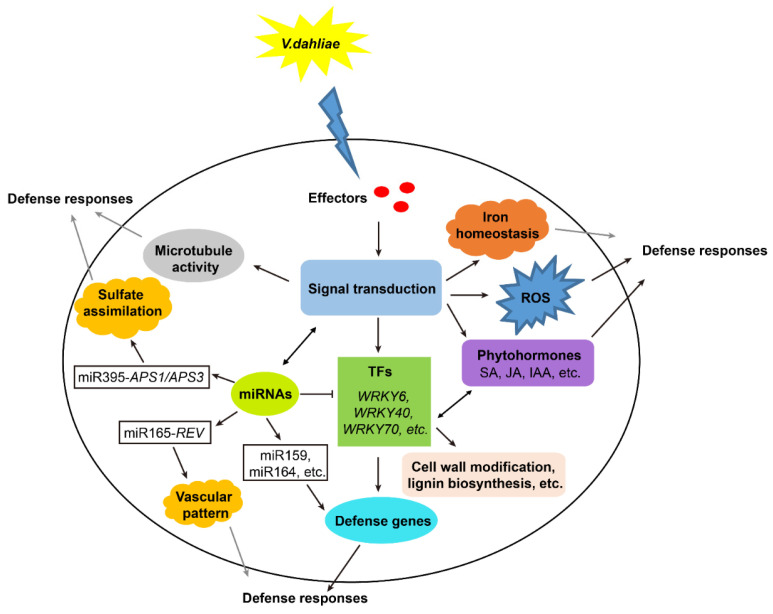
A summary of the putative and verified biological events incurred under *V. dahliae* stress in cotton. Black arrows represent a cascade of verified reaction involving effector proteins, ROS, phytohormones, miRNAs, TFs, etc., which will trigger the plant immunity responses. Grey arrows represent a cascade of putative reaction involving microtubule activity, iron homeostasis, miRNA-targets modules, etc., which may initiate the plant defense responses.

## Data Availability

All the RNA-seq data generated and used in this work have been deposited in Sequence Read Archive (SRA) of the NCBI under the accession numbers PRJNA819185.

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
