# Peer review of "Integrative Analysis of Expression Profiles of mRNA and MicroRNA Provides Insights of Cotton Response to Verticillium dahliae"

_ijms, 2022, doi:10.3390/ijms23094702_

Round 1
Reviewer 1 Report
See the attached file, please

Reviewer 2 Report
The authors have carried out mRNA and miRNA profiling to identify crucial candidates regulating resistance to a fungal pathogen in cotton. The study design is scientifically sound, and the miRNA-target modules identified in this study shall be useful in future to further elucidate the regulatory mechanisms affecting resistance against V. dahliae.
The manuscript is acceptable if some of the concerns are addressed.
A major problem with the manuscript is grammatical errors, which severely impact its readability. I suggest that the authors make use of professional editing services for improvement of the errors. It is not possible to list all the errors. However, few of them are pointed below:
- page 2 line number 43 - 'cotton is one of the......' must be revised to cotton is one of the most important economic crops grown worldwide
- line number 46-47 is unclear - ......which severely restricts cotton productivity and upland cotton
- line number 48- 97% of all cultivated cotton is sensitive - do the authors mean 97% of the cultivars?
- line number 52- there is no doubt that........this phrase is not apt for scientific writing
- line number 53- one of the most effective way.....remains..... is not a good phrase to use here and may possibly be revised to - .........the most effective way...is....
Similarly, there are many issues with the scientific writing, which must be improved.
line number 89 - sliced must be changed to targeted
line number 121 - Q30 ranged from 92.11 to 93.62% - the figures can be rounded of to one decimal place
line number 130 - under J11 and Z2 is not clear
Table 1 - what is the meaning of effective rate? Better terminology can be used here ,does it mean percentage of clean reads? Also, if the number of clean reads is given, there is no need of adding the column for clean base, and the percentage of clean reads can be given along side the number of clean reads, in brackets. Also, the number of genome aligned reads can be added in another column.
Fig 1 - Legend must be revised - Volcano plot instead of volcanic map
Line number 147 - higher proportion of oxidoreductase......instead of proportion, enrichment must be calculated based on hypergeometric test or any other statistical test to determine if these categories are actually enriched. Mere high proportion does not signify that these categories are related to the biological function.
Line number 161 Grammar
Figure 2 - It would be better if the statistical significance of enrichment is also shown here using any kind of scale
line number 171 - instead of many, total number of transcription factor families can be mentioned, along with the percentage of top TF families to better explain the selection of WRKY for further elaboration
line 183 - 4 upregulation?
line 205 - grammar
Figure 4 - two more figures can be added here as parts c and d - one showing the frequency distribution of 330 plus 53 miRNAs based on the length (bp), and another figure detailing the chromosomal distribution of miRNAs in cotton genome
line 224 - grammar
line 228 - higher and lower differential expression is not a correct phrase
line 236 - Here also, it is not clear whether 'most common' terms have been determined based on any statistical test. What is the percentage of target genes mapping to these terms? Fold enrichment would be a better way to actually call them as most common.
figure 8 - What are a. b. c. d?
line 317 -grammar
line 322 - Grammar
In fact, grammar needs to be improved for the entire discussion section
It would be useful to include a conclusive figure which summarizes the most useful functional categories, genes, and miRNA-target modules which are possibly involved in regulating resistance. This would enhance the readability as well as the applicabililty of the manuscript. In case inclusion of this figure exceeds the total number of figures allowed in the manuscript, figure 4 could be moved to supplementary.
Reviewer 3 Report
Dear Authors,
I had great honor to review manuscript entitled: “Integrative analysis of expression profiles of mRNA and microRNA provides insights of cotton response to Verticillium dahliae” which is considered for publication in IJMS. Article is interesting and good written so I suggest minor improvements in a form of list:
- I strongly recommend to shorten the abstract now it is too long
- Introduction must have a precisely formulated aim of the study and hypothesis (as IJMS publication rules follows|). Currently manuscript has nothing like that is present in this section
- Results move Table 1 into supplementary files. I strongly recommend to reduce number of figures in main manuscript. Currently Figures are very complex and in this context 8 of them is to much. I recommend to reduce number of Figures to max 6 rest of Figures must be moved into supplementary Files.
Sincerely,
Round 2
Reviewer 3 Report
Dear Authors,
I recomend publication
Sincerely,